# Factors associated with postoperative visual function after rhegmatogenous retinal detachment with foveal detachment

**Yuki Sugioka[1], Sho Yokoyama**  **[1]\*, Toshio Mori[2], Taisuke Matsuda[1], Tatsushi Kaga[1]**

**1** Department of Ophthalmology, Japan Community Healthcare Organization Chukyo Hospital, Nagoya-City, Aichi, Japan, **2** Department of Ophthalmology, Iida Municipal Hospital, Iida-City, Nagano, Japan

\* yokoyama@chukyogroup.jp

**Data Availability Statement:** All relevant data are within the manuscript and its Supporting Information files.

## Abstract

### Purpose

To investigate pre-, intra-, and postoperative factors influencing postoperative visual acuity, degree of metamorphopsia, and retinal sensitivity after vitrectomy in patients with rhegmatogenous retinal detachment and foveal detachment.

### Methods

We reviewed retrospectively 33 consecutive eyes of 32 patients, who underwent vitrectomy for rhegmatogenous retinal detachment with foveal detachment between August 2018 and October 2020 and obtained retinal reattachment. Pre-, intra-, and postoperative characteristics were comprehensively analyzed using multivariate models to evaluate the presence of factors influencing best-corrected visual acuity, vertical/horizontal metamorphopsia scores using M-CHARTS (Inami & Co., Ltd., Tokyo, Japan), and retinal sensitivity using the MP-3 (NIDEK Co., Aichi, Japan) at 1-year postoperatively.

### Results

Preoperative total retinal detachment was the only factor significantly associated with worse best-corrected visual acuity at 1-year postoperatively (β = 0.589, P<0.001). Intraoperative internal limiting membrane peeling (β = 0.443, P = 0.003) and longer duration after recognizing visual dysfunction (β = 0.425, P = 0.005) were significantly associated with higher vertical metamorphopsia scores at 1 year. The horizontal metamorphopsia score was significantly related to the duration after recognizing visual dysfunction (β = 0.457, P = 0.008). The disappearance of the EZ line on optical coherence tomography at 3 months postoperatively (β = −0.638, P<0.001) was significantly associated with lower retinal sensitivity at 1 year.

### Conclusions

Our study findings suggest that best-corrected visual acuity, metamorphopsia, and retinal sensitivity at 1 year after vitrectomy for rhegmatogenous retinal detachment with foveal detachment are influenced by distinct factors.

**Funding:** The author(s) received no specific funding for this work.

**Competing interests:** The authors have declared that no competing interests exist.

## Introduction

Rhegmatogenous retinal detachment (RRD) is characterized by the intrusion of liquefied vitreous into the space between the sensory retina and retinal pigment epithelium through retinal breaks. If untreated, this condition can lead to eyeball atrophy and blindness. Treatment options include scleral buckling and vitreous surgeries, which aim to close the retinal breaks and achieve retinal reattachment. Recent improvements in the safety of vitreous surgery approaches and the rates of initial retinal reattachment can be attributed to the development of small-incision vitreous surgery, wide-angle fundus viewing systems, and advanced vitreous surgery equipment [1, 2]. Despite successful retinal reattachment and anatomical improvements postoperatively, persistent visual dysfunction remains a challenge [3, 4]. Notably, decreased visual acuity and visual dysfunction, including metamorphopsia and decreased retinal sensitivity, endure for an extended period in cases of RRD with foveal detachment [5]. Therefore, understanding factors contributing to visual dysfunction after RRD surgery is crucial for predicting patient outcomes and enhancing postoperative prognosis. While factors influencing visual acuity after RRD surgery with foveal detachment have been explored [6], there is a dearth of reports investigating factors associated with visual functions such as metamorphopsia and retinal sensitivity. M-CHARTS (Inami & Co., Ltd., Tokyo, Japan) and microperimeters are invaluable tools for the evaluation of macular function. M-CHARTS objectively quantifies the participant's perception of metamorphopsia in both vertical and horizontal directions [7]. The MP-3 (NIDEK Co., Aichi, Japan), the latest generation of microperimetry devices, offers a broader range of stimulus intensities and greater stimulus luminance, enabling the evaluation of low sensitivity [8]. Photopic vision measurement in the brightness field is essential for understanding macular function, particularly cone function. The MP-3 excels in evaluating macular cone function compared to that by conventional microperimeters like MP-1 (NIDEK Co.) and MAIA (Topcon Co., Tokyo, Japan), owing to its capability for brightfield visual field examinations [9]. The MP-3 device also features faster tracking, increased automation, and a broader dynamic range compared to those by the MP-1 [10].

This study aimed to investigate the pre-, intra-, and postoperative factors influencing BCVA, M-CHARTS scores of metamorphopsia, and retinal sensitivity using the MP-3, 1-year postoperatively after vitrectomy in patients with RRD and central foveal detachment.

## Methods

### Study design

This retrospective, observational, cohort study received approval from the Institutional Review Board of Japan Community Healthcare Organization (JCHO) Chukyo Hospital (approval number: 2022015). The study was conducted in adherence to the principles of the 1989 Declaration of Helsinki. The data were accessed for research purposes on 15 July 2022, and all extracted patient data were anonymized for analysis. All patients provided written informed consent and permission to use the clinical data in this study.

### Participants

We retrospectively included in this study 33 consecutive eyes of 32 patients (23 eyes of 22 men, 10 eyes of 10 women), who underwent vitrectomy for RRD with foveal detachment between August 2018 and October 2020 at JCHO Chukyo Hospital and achieved retinal reattachment. Participant characteristics, detailed in Table 1, indicate an average age of 56.8±9.7 (range: 36–78) years. Exclusion criteria comprised eyes with giant tears >90˚, advanced proliferative vitreoretinopathy (PVR) grade C, familial exudative vitreoretinopathy, prior vitreoretinal surgery, eyes with macular

**Table 1. Baseline characteristics of 33 eyes in 32 patients.**

| | |
|---|---|
| **Age, years** | **56.8±9.7 (36 to 78)** |
| **Sex** | |
| **Male** | 23 (69.7) |
| **Female** | 10 (30.3) |
| **Eyes** | |
| **Right** | 16 (48.5) |
| **Left** | 17 (51.5) |
| **Preoperative BCVA, logMAR** | 1.31±0.69 (−0.08 to 2.30)* |
| **Preoperative IOP, mmHg** | 12.03±3.21 (4 to 19)* |
| **Duration after recognizing visual dysfunction, days** | 8.59±21.01 (1 to 120) |
| **Quadrant of retinal detachment area** | |
| **<1** | 1 (3.0) |
| **1–2** | 12 (36.4) |
| **2–3** | 16 (48.5) |
| **3–4** | 1 (3.0) |
| **Total** | 3 (9.1) |
| **PVR grade** | |
| **None** | 15 (45.5) |
| **A** | 8 (24.2) |
| **B** | 10 (30.3) |
| **Surgical procedure** | |
| **PPV and PEA without IOL implantation** | 22 (66.7) |
| **Lens-sparing PPV** | 6 (18.2) |
| **PPV for IOL-implanted eye** | 5 (15.1) |
| **ILM peeling** | |
| **Yes** | 17 (51.5) |
| **No** | 16 (48.5) |
| **Material used for tamponade** | |
| **Air** | 6 (18.2) |
| **SF6** | 24 (72.7) |
| **C3F8** | 3 (9.1) |

Data are presented as mean ± standard deviation (range) or n (%).

* Preoperative BCVA and IOP were analyzed in 32 participants because one participant (No. 7) could not be examined (S1 Table).

BCVA, best-corrected visual acuity; C3F8, perfluoropropane; ILM, internal limiting membrane; IOL, intraocular lens; IOP, intraocular pressure; logMAR, logarithm of the minimum angle of resolution; PEA, phacoemulsification and aspiration; PPV, pars plana vitrectomy; PVR, proliferative vitreoretinopathy; SF6, sulfur hexafluoride.

diseases (including macular hole, epiretinal membrane [ERM], and age-related macular degeneration), diabetic retinopathy, retinal vein occlusion, uveitis, or glaucoma, as these conditions might impact retinal function. Eyes with postoperative ERM, postoperative macular edema, or residual subfoveal fluid after optical coherence tomography (OCT) at 1 week postoperatively, when gas or air occupied less than half of the vitreous cavity, were also excluded.

## Surgical techniques

Pars plana vitrectomy (PPV) and phacoemulsification and aspiration (PEA) were performed in 22 eyes, whereas PPV alone was conducted in 11 eyes (phakic: 6 eyes, intraocular lens [IOL]

implantation: 5 eyes). Lens extraction was performed in cases where it could affect intraoperative fundus visibility. As the subject of this study was RRD with foveal detachment, accurate preoperative measurement of axial length was considered difficult, so IOL implantation was not performed, and IOL insertion was planned at a later date after retinal restoration was obtained in the group that underwent PEA (PPV and PEA without IOL implantation group, 22 eyes, Table 1). All patients underwent endoscope-assisted vitrectomy without scleral buckling similar to the approach used in our previous report [11]. For subretinal fluid drainage, the patient's head was tilted to the position where the retinal break was at its lowest point, and fluid-air exchange was performed under ophthalmic endoscopic observation to drain the fluid through the retinal break. Internal limiting membrane (ILM) peeling was performed in 17 eyes but not in the other 16 eyes. The surgeon decided whether to perform ILM peeling.

## Study outcomes

The primary endpoint of this investigation was the analysis of pre-, intra-, and postoperative factors associated with each of the following outcomes at 1-year postoperatively: best-corrected visual acuity (BCVA), vertical and horizontal metamorphopsia scores (˚) measured with M-CHARTS, and retinal sensitivity (dB) assessed with the MP-3. Additionally, changes in visual acuity, vertical and horizontal metamorphopsia scores, and retinal sensitivity at 3 months, 6 months, and 1-year postoperatively were explored in all 33 study eyes.

## Factors examined in this study

**Preoperative factors.** Age, sex, eye undergoing surgery (right, left), preoperative BCVA, preoperative intraocular pressure (mmHg), and duration after recognizing visual dysfunction (days).

**Intraoperative factors.** Quadrant of retinal detachment area (<1, 1–2, 2–3, 3–4, total), PVR grade (none, A, B), ILM peeling (yes, no), and material used for tamponade (air, sulfur hexafluoride, perfluoropropane).

**Postoperative factors.** Status of the ellipsoid zone (EZ) line at 3 months and 1-year postoperatively (continuous, discontinuous, disappeared).

These factors were assessed to comprehensively understand their influence on the specified visual and anatomical outcomes over the postoperative period.

Preoperative BCVA and IOP were analyzed in only 32 participants because one participant (No. 7) could not be examined (S1 Table). BCVA with decimal visual acuity was converted to the logarithm of the minimum angle of resolution (logMAR) for subsequent analyses. Metamorphopsia was evaluated using M-CHARTS in both vertical and horizontal directions, and the resulting scores were analyzed. The duration after recognizing visual dysfunction (in days) was defined as the period during which the patient acknowledged central vision dysfunction. The MP-3 program employed indicators at 33 points within a 20˚ diameter in the macular region to measure retinal sensitivity (Fig 1). The background brightness was set at 31.4 asb, the stimulus size was Goldman III, the strategy utilized was 4–2 fast, and the fixation point was a 1.0˚ circle. For retinal sensitivity analysis, average values of retinal sensitivity at the 33 points within the 20˚ macular diameter were used. Horizontal and vertical scans through the central fovea were conducted for the EZ line of the central fovea in the OCT scan. The continuity of the EZ within a 1-mm range from the central fovea was assessed. The EZ line was categorized as "continuous" if it exhibited continuity in both horizontal and vertical directions. Any discontinuity in either direction was designated as "discontinuity," whereas disappearance in both horizontal and vertical directions was classified as "disappearance." Representative cases are presented in Fig 2.

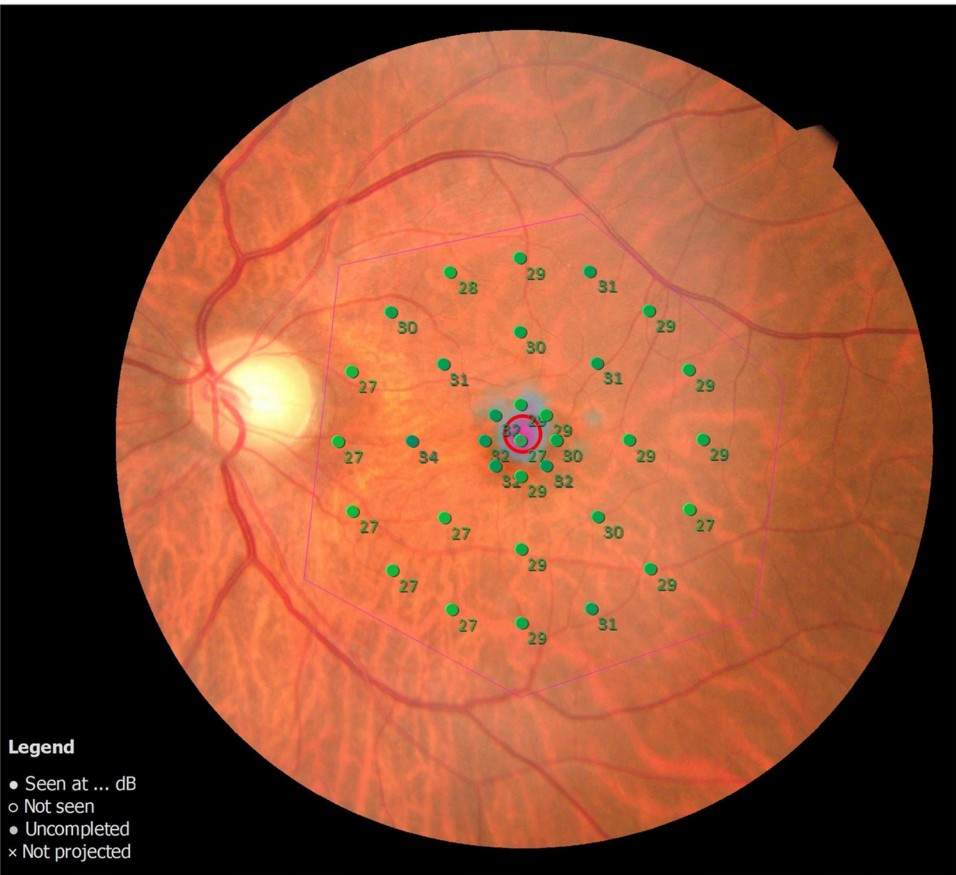

**Fig 1. MP-3 program used for the measurements.** Thirty-three indices within a 20˚ diameter were utilized to assess retinal sensitivity.

## Statistical analysis

Data are presented as mean ± standard deviation or number (percentage). Initially, univariate linear regression analysis was performed to estimate the relationships of pre-, intra-, and postoperative factors with postoperative visual function. Subsequently, multivariate linear regression analyses were conducted using the stepwise method, incorporating factors with P-values <0.1 in the univariate linear regression analysis as explanatory variables and BCVA, vertical and horizontal metamorphopsia degree determined using M-CHARTS, and retinal sensitivity determined using the MP-3 at 1-year postoperatively as the objective variables. For the comparison of postoperative BCVA and metamorphopsia values, Friedman and Dunn's multiple comparison tests were employed. Repeated ANOVA and Tukey's multiple comparison tests were used to compare the average values of retinal sensitivities at 33 points. The Mann–Whitney test was applied for the comparison of BCVA and metamorphopsia values. The Kolmogorov–Smirnov test was used to determine the distribution of variables in each group. A P-value <0.05 was considered statistically significant. Statistical analyses were conducted using SPSS software (version 21; SPSS, Inc., Chicago, IL).

## Results

### Factors related to visual function at 1-year postoperatively

**Linear regression analysis.**    The outcomes of univariate linear regression analysis, as presented in Table 2, delineate the associations between visual function (BCVA,

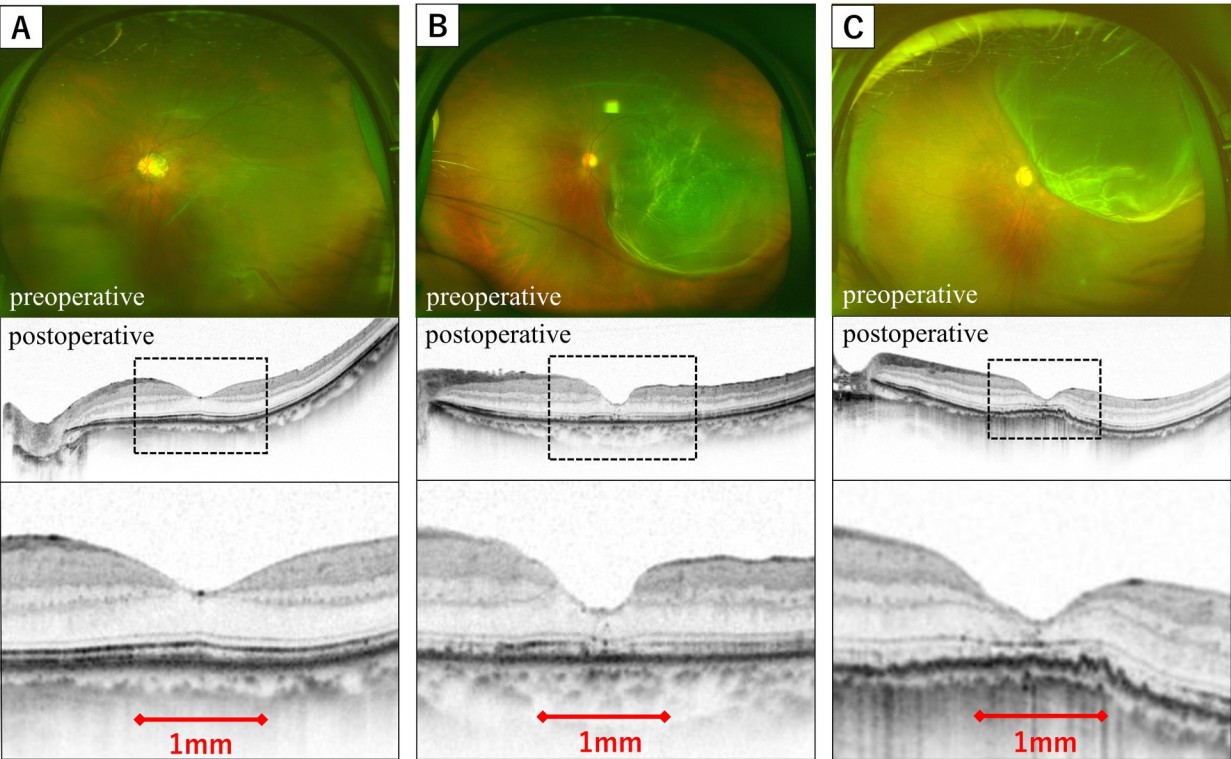

**Fig 2. Representative cases illustrating preoperative fundus photography and postoperative ellipsoid zone (EZ) line status at the central fovea in optical coherence tomography (OCT) scans.** Horizontal and vertical OCT scans through the central fovea were conducted for EZ line assessment to evaluate the continuity of the EZ in the 1- mm range from the central fovea. The displayed OCT images are postoperative horizontal scans through the central fovea, with the lower images representing the magnified areas outlined by black dotted lines in the middle images. A: Continuous EZ line (No. 11). B: Discontinuous EZ line (No. 2). C: Disappeared EZ line (No. 29) (S1 Table).

metamorphopsia, and retinal sensitivity) at 1-year postoperatively and each pre-, intra-, and postoperative factor.

**Multiple regression analysis.** The results of the multiple regression analysis are presented in Table 3. Preoperative total retinal detachment was the only factor significantly associated with worse BCVA at 1-year postoperatively (β = 0.589, P<0.001). For vertical metamorphopsia at 1 year, intraoperative ILM peeling (β = 0.443, P = 0.003) and longer duration after recognizing visual dysfunction (β = 0.425, P = 0.005) were significantly associated with higher scores. The horizontal metamorphopsia score was significantly related to the duration after recognizing visual dysfunction (β = 0.457, P = 0.008), indicating that a longer time to surgery was associated with worse scores for both vertical and horizontal metamorphopsia. Additionally, the disappearance of the EZ line on OCT at 3 months postoperatively (β = −0.638, P<0.001) was significantly associated with lower retinal sensitivity at 1 year.

## Visual function up to 1-year postoperatively

Table 4 presents the BCVA (logMAR), scores of vertical/horizontal metamorphopsia (˚), and average values of retinal sensitivity (dB) at 33 points within a 20˚ diameter in the macular area for all participants at 3 months, 6 months, and 1 year after surgery. BCVA demonstrated a significant improvement in all patients (Friedman test P = 0.002), with postoperative BCVA at 1 year significantly improved compared to the postoperative BCVA at 3 months (Dunn's multiple comparison test P<0.05). However, the scores of vertical and horizontal metamorphopsia

**Table 2. Univariate linear regression analysis of pre-, intra-, and postoperative factors influencing visual function at 1-year postoperatively.**

| Factor | BCVA | | Metamorphopsia score (vertical) | | Metamorphopsia score (horizontal) | | Retinal sensitivity | |
|---|---|---|---|---|---|---|---|---|
| | r | P | r | P | r | P | r | P |
| Age | 0.211 | 0.237 | 0.202 | 0.259 | 0.111 | 0.537 | 0.184 | 0.306 |
| Sex | 0.292 | 0.099 | 0.148 | 0.413 | 0.102 | 0.571 | 0.225 | 0.207 |
| Eyes | 0.208 | 0.246 | 0.054 | 0.767 | 0.205 | 0.253 | 0.078 | 0.665 |
| Preoperative BCVA | 0.421 | *0.016*\* | 0.036 | 0.847 | 0.016 | 0.932 | 0.270 | 0.136 |
| Preoperative IOP | 0.215 | 0.237 | 0.064 | 0.727 | 0.130 | 0.479 | 0.055 | 0.764 |
| Duration after recognizing visual dysfunction | 0.074 | 0.687 | 0.496 | *0.004*\* | 0.467 | *0.007*\* | 0.041 | 0.825 |
| Quadrant of retinal detachment area | | | | | | | | |
| <1 | 0.146 | 0.419 | 0.009 | 0.959 | 0.048 | 0.792 | 0.078 | 0.666 |
| 1–2 | 0.303 | 0.086 | 0.109 | 0.544 | 0.026 | 0.888 | 0.015 | 0.934 |
| 2–3 | 0.087 | 0.628 | 0.037 | 0.836 | 0.094 | 0.602 | 0.202 | 0.261 |
| 3–4 | 0.026 | 0.885 | 0.199 | 0.267 | 0.012 | 0.947 | 0.051 | 0.777 |
| Total | 0.588 | <*0.001*\* | 0.006 | 0.973 | 0.171 | 0.342 | 0.359 | *0.040*\* |
| PVR grade | | | | | | | | |
| None | 0.306 | 0.083 | 0.292 | 0.099 | 0.111 | 0.539 | 0.105 | 0.560 |
| A | 0.144 | 0.422 | 0.046 | 0.797 | 0.095 | 0.597 | 0.027 | 0.881 |
| B | 0.466 | *0.006*\* | 0.360 | *0.040*\* | 0.209 | 0.243 | 0.139 | 0.439 |
| Presence of ILM peeling | 0.156 | 0.386 | 0.496 | *0.003*\* | 0.360 | *0.040*\* | 0.197 | 0.272 |
| EZ at 3 months postoperatively | | | | | | | | |
| Continuous | 0.342 | 0.051 | 0.395 | *0.023*\* | 0.268 | 0.132 | 0.050 | 0.784 |
| Discontinuous | 0.086 | 0.633 | 0.348 | *0.047*\* | 0.303 | 0.087 | 0.508 | *0.003*\* |
| Disappeared | 0.494 | *0.004*\* | 0.040 | 0.823 | 0.119 | 0.509 | 0.638 | <*0.001*\* |
| EZ at 1-year postoperatively | | | | | | | | |
| Continuous | 0.317 | 0.072 | 0.258 | 0.148 | 0.048 | 0.792 | 0.359 | *0.040*\* |
| Discontinuous | 0.317 | 0.072 | 0.258 | 0.148 | 0.048 | 0.792 | 0.359 | *0.040*\* |

\* Statistically significant.

BCVA, best-corrected visual acuity; EZ, ellipsoid zone; ILM, internal limiting membrane; IOP, intraocular pressure; PVR, proliferative vitreoretinopathy.

and the average values of retinal sensitivity at 33 points within a 20˚ diameter in the macular area did not significantly improve (P = 0.253, 0.211, and 0.223, respectively).

## Discussion

In this study investigating factors related to visual function at 1-year postoperatively, we found that preoperative total retinal detachment was significantly associated with worse BCVA after

**Table 3. Stepwise multivariate regression linear regression analysis of pre-, intra-, and postoperative factors affecting visual function at 1-year postoperatively.**

| | Related factors | β | P-value |
|---|---|---|---|
| BCVA | Preoperative total retinal detachment | 0.589 | <*0.001*\* |
| Metamorphopsia score (vertical) | Presence of ILM peeling | 0.443 | *0.003*\* |
| | Duration after recognizing visual dysfunction | 0.425 | *0.005*\* |
| Metamorphopsia score (horizontal) | Duration after recognizing visual dysfunction | 0.457 | *0.008*\* |
| Retinal sensitivity | Discontinuity of the EZ line at 3 months postoperatively | −0.638 | <*0.001*\* |

\* Statistically significant.

BCVA, best-corrected visual acuity; EZ, ellipsoid zone; ILM, internal limiting membrane.

**Table 4. Postoperative outcomes regarding visual acuity, metamorphopsia, and retinal sensitivity in all participants.**

|  |  | 3 months | 6 months | 1 year | P-value |
|---|---|---|---|---|---|
| **BCVA (logMAR)** |  | 0.16±0.25 | 0.14±0.29 | 0.09±0.26 | ***0.002*** * |
| **Metamorphopsia score (˚)** | **Vertical** | 0.69±0.54 | 0.59±0.45 | 0.58±0.47 | 0.211 |
|  | **Horizontal** | 0.64±0.51 | 0.58±0.57 | 0.53±0.50 | 0.223 |
| **Retinal sensitivity (dB)** |  | 25.60±1.60 | 25.92±1.95 | 25.85±1.95 | 0.253 |

Data are presented as mean ± standard deviation.

* Statistically significant.

BCVA, best-corrected visual acuity; logMAR, logarithm of the minimal angle of resolution.

surgery. Intraoperative ILM peeling and longer duration after recognizing visual dysfunction were significantly associated with higher vertical metamorphopsia scores, and the horizontal metamorphopsia score was significantly related to the duration after recognizing visual dysfunction. Additionally, the disappearance of the EZ line on OCT at 3 months postoperatively was significantly associated with lower retinal sensitivity.

Park et al. examined factors influencing visual acuity 1-year postoperatively in 180 eyes undergoing surgery for RRD with foveal involvement. They reported that predictive factors included the extent of detachment, duration of macula-off, preoperative external limiting membrane (ELM) integrity, and postoperative outer retinal microstructures, particularly the photoreceptor outer segment layer [6]. Additionally, previous research has emphasized the significance of OCT morphology after RRD surgery, indicating that the continuity of the postoperative EZ and ELM lines, as well as the presence of a foveal bulge, correlate with final BCVA after RRD surgery [12–14].

On the other hand, our study identified total detachment as a significant factor related to BCVA 1-year postoperatively, aligning with Park et al.'s findings that preoperative predictive factors include the extent of detachment. However, our study did not identify the duration after recognizing visual dysfunction or the postoperative status of the EZ line as associated factors. The discrepancy between previous reports of the duration of macula-off and the OCT morphology after RRD surgery being associated with postoperative visual acuity and our results may be attributed to the relatively small sample size in our study and the less stringent evaluation of the duration of foveal detachment.

The duration after recognizing visual dysfunction was identified as an associated factor for both vertical and horizontal metamorphopsia scores assessed using M-CHARTS. Although the duration after recognizing visual dysfunction may not always coincide with the period of foveal detachment, these two intervals are considered similar owing to the impact of central foveal detachment on visual acuity. Thereby, the current study results suggest that postoperative metamorphopsia may worsen with a longer duration since the patient acknowledged central vision dysfunction.

Additionally, intraoperative ILM peeling was associated with worse postoperative 1-year vertical metamorphopsia scores. In the context of vitrectomy for patients with RRD, ILM peeling may be performed to prevent postoperative ERMs [15, 16]. In regard to postoperative retinal function with ILM peeling in RRD cases, Pietras-Trzpiel et al. reported no significant difference in postoperative metamorphopsia scores, assessed using M-CHARTS between patients with or without ILM peeling during vitrectomy for RRD [17]. Conversely, Abdullah et al. evaluated macular structure and function with and without ILM peeling for RRD with foveal detachment using OCT angiography and multifocal electroretinography, revealing potential damage to macular structure and function in the ILM peeling group [18]. However,

in that study, perfluorocarbon liquids (PFCLs) and silicone oil (SO) tamponade were used in all patients. Reportedly, the use of PFCLs is associated with increased discontinuity in the interdigitation zone following RRD surgery compared with that when not using PFCLs [19]. Additionally, SO tamponade has been associated with postoperative thinning of both the inner and outer retinal layers compared with that using gas tamponade [20]. Therefore, it is important to consider the potential effects of PFCLs and SO tamponade on macular structure and function.

In the current study, the decision to perform ILM peeling was left to the discretion of individual surgeons, and it is possible that more severe cases were more likely to undergo ILM peeling. Indeed, the higher the preoperative PVR grade, the more frequently ILM peeling was performed. However, the results of multiple regression analysis, which included the presence of ILM peeling and PVR grade as factors, showed that PVR grade was not related to postoperative visual function, including visual acuity, metamorphopsia, and retinal sensitivity. In summary, the current study suggests that ILM peeling may impact retinal function and exacerbate metamorphopsia regardless of the severity of the PVR grades.

Retinal sensitivity measurements demonstrated an association with the morphology of the EZ line at 3-months postoperatively, implying that patients in whom the EZ line in the central fovea disappeared at 3-months postoperatively might experience reduced retinal sensitivity within the central 20˚ macular area at 1-year postoperatively. The EZ line at 3-months postoperatively was continuous in five patients, discontinuous in 24 patients, and absent in four patients. In contrast, the EZ line at 1-year postoperatively was continuous in 22 patients and discontinuous in 11 patients, with no case of disappearance, indicating an improvement in the EZ line over time after surgery. Thus, the initial state of the EZ line at 3 months is important for predicting long-term retinal sensitivity, although many patients experience recovery of the EZ line from 3 months-to 1-year postoperatively.

In our study, only visual acuity showed significant improvement, with postoperative BCVA being significantly better at 1 year than at 3 months after surgery. However, the scores for vertical/horizontal metamorphopsia and the average values of retinal sensitivity in the macula showed no significant improvement between 3 months and 1 year after surgery. This suggests that the degree of metamorphopsia and retinal sensitivity may not significantly improve in patients with RRD and foveal detachment following surgery. Several studies have investigated the postoperative visual function in patients with RRD and foveal detachment. Borowicz et al. reported significant improvements in BCVA, retinal sensitivity, and metamorphopsia between 1-and 6-months postoperatively [21]. Yamada et al. [22] and Okuda et al. [23] evaluated BCVA and metamorphopsia from 1 to 12 months after surgery, noting a significant improvement in metamorphopsia, although BCVA did not improve significantly (Table 5).

The greatest discrepancy in postoperative changes between previous reports and the present study is with regard to metamorphopsia. The metamorphopsia score improved significantly over time in all other reports including Borowicz et al. [21], Yamada et al. [22], and Okuda et al. [23]; however, no improvement was observed in the present study. Moreover, retinal sensitivity improved over time in Borowicz et al. [21], whereas no such improvement was observed in our study.

Several reasons might account for these differences. First, the preoperative RRD status and the duration of foveal detachment might have varied between previous studies and the current study. Yamada et al. [22] reported a period from subjective symptoms to surgery of 5.25±6.37 days, which is shorter than the 8.59±21.01 days observed in our study. Moreover, the retinal detachment area affected fewer than two quadrants in 27 out of 33 cases in Yamada et al., suggesting that the preoperative retinal detachment status in their study might have been milder than that in the current study. Additionally, quadrant of retinal detachment area and the

**Table 5. Comparison of studies examining visual function after surgery for rhegmatogenous retinal detachment with foveal detachment.**

| | | Pre | 1 month | 3 months | 6 months | 1 year | Significant difference |
|---|---|---|---|---|---|---|---|
| **Present study, 2024** (n = 33) | BCVA (logMAR) | 1.31±0.69* | — | 0.16±0.25 | 0.14±0.29 | 0.09±0.26 | _Yes_ |
| | Metamorphopsia score | — | — | V: 0.69±0.54 H: 0.64±0.51 | V: 0.59±0.45 H: 0.58±0.57 | V: 0.58±0.47 H: 0.53±0.50 | No |
| | Retinal sensitivity (MP-3) | — | — | 25.60±1.60 | 25.92±1.95 | 25.85±1.95 | No |
| **Borowicz et al., 2019** [21] (n = 28) | BCVA (logMAR) | 1.70±0.18 | — | — | 0.40±0.28 | — | _Yes_ |
| | Metamorphopsia score | — | V: 0.20±0.30 H: 0.20±0.41 | V: 0.10±0.25 H: 0.00±0.34 | V: 0.00±0.25 H: 0.00±0.31 | — | _Yes_ |
| | Retinal sensitivity (MAIA) | — | 23.80±4.41 | 25.10±3.53 | 26.10±4.18 | — | _Yes_ |
| **Yamada et al., 2021** [22] (n = 33) | BCVA (logMAR) | 0.92±0.61 | 0.15±0.21 | 0.07±0.16 | 0.03±0.16 | −0.02±0.13 | No |
| | Metamorphopsia score | — | 0.71±0.41 | 0.66±0.45 | 0.62±0.44 | V: 0.45±0.47 H: 0.51±0.57 | _Yes_ |
| **Okuda et al., 2018** [23] (n = 26) | BCVA (logMAR) | 1.10±0.59 | 0.17±0.24 | — | 0.02±0.12 | 0.00±0.12 | No |
| | Metamorphopsia score | — | V: 0.73±0.52 H: 0.60±0.37 | — | V: 0.48±0.44 H: 0.40±0.42 | V: 0.29±0.34 H: 0.23±0.25 | _Yes_ |

* Preoperative BCVA in the present study was analyzed in 32 participants because one participant (No. 7) could not be examined (S1 Table).

BCVA, best-corrected visual acuity; H, horizontal; logMAR, logarithm of the minimal angle of resolution; V, vertical.

duration of foveal detachment were not specified in Borowicz et al. [21] and Okuda et al. [23]. Therefore, the recovery over time might have differed among these four studies depending on the retinal detachment status of the patients. Second, in all previous reports [21–23], the examination was performed at 1-month postoperatively, but in the current study, the examination was performed only from 3-months postoperatively. Therefore, recovery of macular function might have been largely completed at 3-months postoperatively, as a result, no further recovery over time was observed. Third, differences in surgical procedures may be related to variability in results. In our study, ILM staining was performed with brilliant blue G instead of indocyanine green, which is known for its retinal toxicity. Additionally, retinotomy was not performed for the drainage of subretinal fluid, which was properly drained under ophthalmic endoscopy by changing the head position. Reportedly, avoiding the creation of a retinotomy during the drainage of subretinal fluid can improve the success rate of RRD surgery and reduce the incidence of postoperative ERM [24, 25]. Additionally, the use of PFCLs has been associated with a reduction in postoperative visual acuity and the disruption of the outer retinal structures [26]. These differences in surgical methods may have favored the early recovery of macular function in the present study, and at 3-months postoperatively, macular function recovery was nearly complete, with no further recovery observed over time.

In the findings of our present study, the factors influencing BCVA, metamorphopsia scores, and retinal sensitivity 1 year after RRD surgery demonstrated disparities, indicating that the recovery trajectories of individual visual functions were not in parallel. Human visual perception encompasses spatial characteristics, temporal characteristics, and color vision. Visual acuity is defined as the spatial discrimination ability and is quantified by the minimum visual angle required to distinguish between two separate objects [27]. M-CHARTS, a measure of visual variability, is believed to assess the disorder in the arrangement of photoreceptor cells. Additionally, retinal sensitivity is considered to gauge photosensitivity, reflecting the ability to sense light. Given that each of these metrics evaluates distinct visual functions, the associated factors and the postoperative recovery processes are presumed to differ.

Limitations of this study are its small sample size and the absence of data in the early postoperative period. Future research should aim to expand the sample size and monitor the

progression over time. It is anticipated that the postoperative recovery process of macular morphology and visual function differs between eyes with high myopia and those without. We should have investigated whether the presence of high myopia influences postoperative visual function. However, the study population included patients with unknown axial lengths, thereby preventing such evaluation. This study included eyes with IOL insertion and phakic eyes, and patients who underwent vitrectomy in phakic eyes did not exhibit noticeable cataract progression at the 1-year postoperative follow-up. Nevertheless, it is necessary to investigate the differences in MP-3 test values between eyes with IOL insertion and phakic eyes, considering the findings of the current study.

In conclusion, the factors influencing postoperative BCVA, metamorphopsia scores, and retinal sensitivity 1 year after surgery in RRD with foveal detachment were identified as follows: BCVA was associated with preoperative total retinal detachment, vertical metamorphopsia score was linked to intraoperative ILM peeling and the duration after recognizing visual dysfunction, horizontal metamorphopsia score was correlated with the duration after recognizing visual dysfunction, and retinal sensitivity was influenced by the disappearance of the EZ line on OCT at 3-months postoperatively. Additionally, visual acuity at 1-year postoperatively showed significant improvement compared to that at 3-months postoperatively. However, metamorphopsia scores and retinal sensitivity did not significantly improve. These findings suggest that factors influencing visual acuity, metamorphopsia, and retinal sensitivity after RRD surgery are distinct, and the recovery process in visual function over time after surgery is not uniform.

## Supporting information

**S1 Table. Preoperative, intraoperative, and postoperative data of 33 rhegmatogenous retinal detachment eyes with foveal detachment.**
(XLSX)

## Acknowledgments

The authors thank Hayato Mitamura, MD, Hiroyuki Sato, MD, and Kazuo Ichikawa, MD, PhD for their help with the manuscript. The authors would also like to thank Editage (www.editage.com) for English language editing.

## Author Contributions

**Conceptualization:** Yuki Sugioka, Sho Yokoyama.

**Data curation:** Yuki Sugioka, Toshio Mori.

**Formal analysis:** Yuki Sugioka.

**Investigation:** Yuki Sugioka.

**Methodology:** Sho Yokoyama.

**Supervision:** Sho Yokoyama, Taisuke Matsuda, Tatsushi Kaga.

**Visualization:** Yuki Sugioka.

**Writing – original draft:** Yuki Sugioka.

**Writing – review & editing:** Sho Yokoyama, Toshio Mori.

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
