## [Decision Letter · Decision Letter 0]

8 Apr 2024

PONE-D-24-04377Factors associated with postoperative visual function after rhegmatogenous retinal detachment with foveal detachmentPLOS ONE

Dear Dr. yokoyama,

Thank you for submitting your manuscript to PLOS ONE. After careful consideration, we feel that it has merit but does not fully meet PLOS ONE’s publication criteria as it currently stands. Therefore, we invite you to submit a revised version of the manuscript that addresses the points raised during the review process. Please submit your revised manuscrip by May 23 2024 11:59PM. If you will need more time than this to complete your revisions, please reply to this message or contact the journal office at plosone@plos.org. Please include the following items when submitting your revised manuscript:A rebuttal letter that responds to each point raised by the academic editor and reviewer(s). You should upload this letter as a separate file labeled 'Response to Reviewers'.A marked-up copy of your manuscript that highlights changes made to the original version. You should upload this as a separate file labeled 'Revised Manuscript with Track Changes'.An unmarked version of your revised paper without tracked changes. You should upload this as a separate file labeled 'Manuscript'.

We look forward to receiving your revised manuscript.

Kind regards,

Jiro Kogo

Academic Editor

PLOS ONE

Journal Requirements:

Reviewers' comments:

Reviewer's Responses to Questions

**Comments to the Author**

1. Is the manuscript technically sound, and do the data support the conclusions?

Reviewer #1: Partly

Reviewer #2: Yes

2. Has the statistical analysis been performed appropriately and rigorously? 

Reviewer #1: I Don't Know

Reviewer #2: Yes

3. Have the authors made all data underlying the findings in their manuscript fully available?

Reviewer #1: Yes

Reviewer #2: Yes

4. Is the manuscript presented in an intelligible fashion and written in standard English?

Reviewer #1: Yes

Reviewer #2: No

5. Review Comments to the Author

Reviewer #1: 1. On Page 9, Line 85, you mentioned“33 eyes of 32 patients”, which is not consistent with “the 33 eyes of 34 patients”mentioned in the methods in the abstract. Please explain? If so, I suggest that it is better to include 32 eyes of 32 people and only one eye of each patient to avoid interference.

2. On Page 9, Line 89, The exclusion criteria did not include high myopia, but did this study include high myopia? After all, the postoperative macular morphology and visual function recovery of high myopia and non-high myopia are different.

3. On Page 14, Line 135-136, you state that "The duration after recognizing visual dysfunction (in days) was defined as the period during which the patient acknowledged central vision dysfunction" this is only true if the macula is involved. Please correct this.

4. On Page 19, Line 178, It would be helpful to describe in the manuscript and tables how many eyes were followed up at 3 months, 6 months, and 12 months. These images are good in the manuscript, but it would also be helpful to have a preoperative photograph.

Reviewer #2: In this article, authors try to reveal factors related to the visual outcomes after repair of macula-involving RRDs including the analyses of retinal sensitivity, and compare their results with other previous papers. Their methods sound reasonable and the results are within expectations, however, I would like to propose several points for further revision particularly for more distinct expression of the results and for discussion to obtain relevance for their results that are not necessarily consistent with the previous ones. Additionally, I would like to ask authors to revise the text to remove descriptions duplicating in two or more sections such as the same thing expressed in the method and the result.

1. For evaluation of visual outcomes after repair of macula-involving RRD, methods of SRF drainage and use of dye are important factors since some previous studies have reported differences in the visual outcomes or morphology of the ORLs depending on the use of PFCL, SO, ICG, or creation of an intentional break for drainage. Please describe these aspects in details in the method as well as in the discussion particularly in comparison with other studies presented in Table 5.

2. Abstract: Methods “33 eyes of 34 patients” should be a mistake.

3. Abstract, the 1st line of the conclusions “The factors related to….. differ”: it is not clear what this sentence means. I guess this sentence could be deleted.

4. Results are presented for both univariate analyses and multiple regression analyses in three paragraphs. They might be understood better if subtitles for each type of analyses are provided, such as ‘Univariate linear regression analysis’ and ‘Multiple regression analysis’. Plus, as a rule of those statistical methods, results of the univariate analysis would be better presented briefly in a very short paragraph because they only serve to show which of the factors were selected for the multiple regression analysis (Table 2 shows everything clearly without explanation in the main text) and authors should concentrate on the results of the multivariate analysis. Otherwise, the descriptions of the results are quite long and complicated and may give readers a wrong message that the results of the univariate analyses have some important meaning.

5. Association of each factor with BCVA, metamorphopsia scores and retinal sensitivity is presented with p-values, but it is not clear whether the association was positive or negative. Please present each association specifically as worse or better visual function. As far as I understood, each factor seems to be related to ‘worse’ BCVA, ‘higher’ metamorphopsia scores and ‘lower’ retinal sensitivity. To help readers understand these aspects correctly, I would also like to request authors to provide the mean values (with SD) of each visual function regarding each factor (at least for the factors that were proved to affect visual function in the multivariate analysis as shown in Table 3) in the table in addition to r and P values.

6. Among the factors proved to be associated with postoperative visual function, ILM peeling is the only one which surgeons can decide to do or not to do. In this meaning, the result regarding ILM peeling is the most suggestive point in this article. Although it was performed on the surgeons’ discretion, I guess there might have been some trend for the decision of ILM peeling such as the time when PPV was performed or the distribution/chronicity of the RRD. Please provide some information and comment on this aspect.

7. In the 1st paragraph of the discussion, a result from univariate analysis is mentioned as being consistent with the result presented by Park et al; however, as a principle and as proposed above in the comment 4, the factors which did not have significance in the multivariate analysis should not be considered as a meaningful result even if significance was shown in the univariate regression analysis.

8. One of the important aspects of this study is the difference from other studies in terms of the postoperative recovery of metamorphopsia and retinal sensitivity as presented in Table 5. I would like to ask authors to add some interpretations and comments to explain these discrepancies. Without such discussions, the results presented in this article might lose its relevance. One possible aspect to explain the discordance might be related to those as suggested in the comment No 1 above.

6. PLOS authors have the option to publish the peer review history of their article (what does this mean?). If published, this will include your full peer review and any attached files.

Reviewer #1: No

Reviewer #2: No

---

## [Author Response · Author response to Decision Letter 0]

19 May 2024

May 19th, 2024

Kogo Jiro 

Academic Editor

PLOS ONE

Re: Revised manuscript, PONE-D-24-04377

Title: Factors associated with postoperative visual function after rhegmatogenous retinal detachment with foveal detachment

Dear Dr. Kogo:

The manuscript has been carefully rechecked, and appropriate changes have been made in accordance with the reviewers’ suggestions. We provide below a point-by-point response to your and the reviewers’ comments. All changes to our manuscript are indicated in red text.

We thank you and the reviewers for your thoughtful suggestions and insights, from which our manuscript has greatly benefited. We look forward to working with you and the reviewers to move this manuscript closer to publication in PLOS ONE.

Thank you for your consideration. I look forward to hearing from you.

Sincerely,

Sho Yokoyama

Department of Ophthalmology, Japan Community Healthcare Organization Chukyo Hospital

1-1-10 Sanjo Minami-ku Nagoya-city, Aichi prefecture, Japan

E-mail: yokoyama@chukyogroup.jp

Dear Editor,

We have submitted all raw dates as a Supporting Information file (Supporting information table 1_pre- and postoperative data of 33 RRD eyes with foveal detachment. Among 33 subjects, preoperative BCVA and IOP were analyzed in 32 subjects because one subject could not be examined.

This was clearly stated within the manuscript (Lines 99-100, 139-140, 294.)

REVIEWER 1 

We thank the reviewer for the thoughtful comments that increased the scientific value of our paper. 

Comments to the Author

Comment 1:

On Page 9, Line 85, you mentioned“33 eyes of 32 patients”, which is not consistent with “the 33 eyes of 34 patients”mentioned in the methods in the abstract. Please explain? If so, I suggest that it is better to include 32 eyes of 32 people and only one eye of each patient to avoid interference.

Response 1:

Thank you for pointing out this inconsistency. In the Abstract, we corrected the number to 33 eyes of 32 patients. As the reviewer mentioned, it is preferable to include one eye per patient. However, two eyes of one patient were included in this study because the protocol was to include patients as consecutive RRD cases during the study period. We have added a statement that this study included consecutive RRD cases (lines 24 and 84).

Comment 2:

On Page 9, Line 89, The exclusion criteria did not include high myopia, but did this study include high myopia? After all, the postoperative macular morphology and visual function recovery of high myopia and non-high myopia are different.

Response 2:

Since the participants in this study were RRD patients with preoperative foveal detachment, the preoperative refractive values cannot be properly evaluated. Of the 27 eyes for which ocular axis length was measured postoperatively, 16 eyes (59%) had high myopia, with an ocular axis length of 26 mm or longer. The remaining 11 eyes did not have high myopia, with an axial length of 22–26 mm. As stated by the reviewer, whether a patient has high myopia is considered an important factor and should be included in the considerations. However, six eyes of the included patients had unknown axis lengths, so we were unable to examine this factor in our study. We discuss as one of the limitations that the presence of high myopia was not analyzed in this study and should be included as an item for consideration in future studies.

The following text has been added to the limitations paragraph.

“It is anticipated that the postoperative recovery process of macular morphology and visual function differs between eyes with high myopia and those without. We should have investigated whether the presence of high myopia influences postoperative visual function. However, the study population included patients with unknown axial lengths, thereby preventing such evaluation.” (lines 326–330)

Comment 3:

On Page 14, Line 135-136, you state that "The duration after recognizing visual dysfunction (in days) was defined as the period during which the patient acknowledged central vision dysfunction" this is only true if the macula is involved. Please correct this.

Response 3:

Only cases with preoperative foveal detachment were included in this study.

Comment 4:

On Page 19, Line 178, It would be helpful to describe in the manuscript and tables how many eyes were followed up at 3 months, 6 months, and 12 months. These images are good in the manuscript, but it would also be helpful to have a preoperative photograph.

Response 4:

All 33 eyes included in the study were examined at 3, 6, and 12 months postoperatively. We added this sentence to the Results section (line 126). 

However, we have difficulties interpreting the reviewer’s comment “These images are good in the manuscript, but it would also be helpful to have a preoperative photograph.”. We understand that this comment refers to Figure 2. Therefore, we added preoperative photographs to Figure 2 and revised the corresponding figure legend (lines 159–166). 

Reviewer #2: In this article, authors try to reveal factors related to the visual outcomes after repair of macula-involving RRDs including the analyses of retinal sensitivity, and compare their results with other previous papers. Their methods sound reasonable and the results are within expectations, however, I would like to propose several points for further revision particularly for more distinct expression of the results and for discussion to obtain relevance for their results that are not necessarily consistent with the previous ones. Additionally, I would like to ask authors to revise the text to remove descriptions duplicating in two or more sections such as the same thing expressed in the method and the result.

REVIEWER 2 

We thank the reviewer for the thoughtful comments that increased the scientific value of our paper. 

Comment 1:

1. For evaluation of visual outcomes after repair of macula-involving RRD, methods of SRF drainage and use of dye are important factors since some previous studies have reported differences in the visual outcomes or morphology of the ORLs depending on the use of PFCL, SO, ICG, or creation of an intentional break for drainage. Please describe these aspects in details in the method as well as in the discussion particularly in comparison with other studies presented in Table 5.

Response 1:

In this study, PFCL was used intraoperatively in 8 (24%) of the 33 eyes, all of which were used to reduce the intraoperative mobility of the detached retina, and none of which were used during drainage. SO was used in none of the cases. BBG was used for membrane staining in all 17 cases in which ILM peeling was performed, and ICG was used in none of the cases. During subretinal fluid (SRF) drainage, all cases were drained through the original retinal breaks, and no cases were drained by creating a drainage retinotomy for SRF drainage or using a PFCL. Although not mentioned in the previous manuscript, all patients underwent endoscope-assisted vitrectomy without scleral buckling. When performing SRF drainage, the patient’s head was tilted to the position where the retinal break was at the lowest position. The SRF was then drained through the retinal break using fluid-air exchange under ophthalmic endoscopic observation. This method is unique to endoscopy and allows to tilt the head position more than under the microscope, for a firm drainage despite not using a PFCL, even if retinal breaks are present in the peripheral retina (Yokoyama S, Kojima T, Mori T, Matsuda T, Sato H, Yoshida N, et al. Clinical outcomes of endoscope-assisted vitrectomy for treatment of rhegmatogenous retinal detachment. Clin Ophthalmol. 2017;11: 2003-2010. https://doi.org/10.2147/OPTH.S147690). The various results of this study that differ from those of previous studies might be related to the fact that ocular endoscopic drainage may have affected postoperative visual function. 

In previous studies evaluating postoperative visual function in RRD with central foveal detachment, no drainage retinotomy was created in the report by Yamada et al. (2021) and Borowicz et al. (2019) and Okuda et al. (2018) were unknown. In addition, the presence or absence of ILM peeling is not described in the reports by Yamada et al. and Okuda et al., whereas ILM peeling was not performed in the report by Borowicz et al.

The most striking difference between the previous studies and the present study was the result regarding metamorphopsia. Yamada et al., Okuda et al., and Borowicz et al. all reported significant improvement in metamorphopsia scores over time, whereas the present study found no such improvement. Moreover, in Borowicz et al., retinal sensitivity improved over time, but no such improvement was detected in the present study. 

Several reasons might account for these differences. First, postoperative visual functions may have been influenced by the preoperative RRD grade and the duration of foveal detachment, and the recovery over time may vary depending on the RRD status of the patient. The preoperative RRD status of participants might have differed among studies. Second, in all previous reports, the examination was performed at 1 month postoperatively, but in the current study, the examination was performed only from 3 months postoperatively. Therefore, it is possible that recovery of macular function was largely complete at 3 months postoperatively so that no further recovery over time was observed. Third, differences in surgical procedures may be related to variability in results. In our study, ILM staining was performed with brilliant blue G instead of indocyanine green, which is known for its retinal toxicity. Additionally, retinotomy was not performed for the drainage of subretinal fluid, which was properly drained under ophthalmic endoscopy by changing the head position. Thus, recovery of macular function may have been nearly complete at 3 months postoperatively, and no further recovery over time was observed. 

These considerations have been added to the Methods and Discussion sections of the revised manuscript.

Methods:

“All patients underwent endoscope-assisted vitrectomy without scleral buckling similar to the approach used in our previous report [11]. For subretinal fluid drainage, the patient's head was tilted to the position where the retinal break was at its lowest point, and fluid-air exchange was performed under ophthalmic endoscopic observation to drain the fluid through the retinal break.” (lines 114–118) 

Discussion:

“The greatest discrepancy in postoperative changes between previous reports and the present study is with regard to metamorphopsia. The metamorphopsia score improved significantly over time in all other reports including Borowicz et al. [19], Yamada et al. [20], and Okuda et al. [21], but no improvement was observed in the present study. Moreover, retinal sensitivity improved over time in Borowicz et al. [19], whereas no such improvement was observed in our study. Several reasons might account for these differences. First, postoperative visual functions may be influenced by the preoperative RRD grade and the duration of foveal detachment, and the recovery over time may vary depending on the RRD status of the patient. The preoperative RRD status of participants might have varied among studies. Second, in all previous reports [19-21], the examination was performed at 1 month postoperatively, but in the current study, the examination was performed only from 3 months postoperatively. Therefore, recovery of macular function might have been largely completed at 3 months postoperatively so that no further recovery over time was observed. Third, differences in surgical procedures may be related to variability in results. In our study, ILM staining was performed with brilliant blue G instead of indocyanine green, which is known for its retinal toxicity. Additionally, retinotomy was not performed for the drainage of subretinal fluid, which was properly drained under ophthalmic endoscopy by changing the head position. These differences in surgical procedures may have led in our study to results that diverge from those of previous studies.” (lines 296–313) 

Comment 2: Abstract: Methods “33 eyes of 34 patients” should be a mistake.

Response 2:

Thank you for pointing this out inconsistency. The Abstract has been revised to 33 eyes of 32 patients. (line 24)

Comment 3: Abstract, the 1st line of the conclusions “The factors related to….. differ”: it is not clear what this sentence means. I guess this sentence could be deleted.

Response 3:

Thank you for this pertinent suggestion. This sentence has been removed.

Comment 4: Results are presented for both univariate analyses and multiple regression analyses in three paragraphs. They might be understood better if subtitles for each type of analyses are provided, such as ‘Univariate linear regression analysis’ and ‘Multiple regression analysis’. Plus, as a rule of those statistical methods, results of the univariate analysis would be better presented briefly in a very short paragraph because they only serve to show which of the factors were selected for the multiple regression analysis (Table 2 shows everything clearly without explanation in the main text) and authors should concentrate on the results of the multivariate analysis. Otherwise, the descriptions of the results are quite long and complicated and may give readers a wrong message that the results of the univariate analyses have some important meaning.

Response 4:

Thank you for the valuable comments, and we agree with the reviewer’s suggestions. The results of the univariate analysis are now stated briefly, as they only indicate which factors were selected for the multiple regression analysis. Moreover, the structure of the Results section has been revised based on your suggestion. 

Comment 5: Association of each factor with BCVA, metamorphopsia scores and retinal sensitivity is presented with p-values, but it is not clear whether the association was positive or negative. Please present each association specifically as worse or better visual function. As far as I understood, each factor seems to be related to ‘worse’ BCVA, ‘higher’ metamorphopsia scores and ‘lower’ retinal sensitivity. To help readers understand these aspects correctly, I would also like to request authors to provide the mean values (with SD) of each visual function regarding each factor (at least for the factors that were proved to affect visual function in the multivariate analysis as shown in Table 3) in the table in addition to r and P values.

Response 5:

As described by you, each factor is associated with "worse" BCVA (logMAR), "higher" metamorphopsia scores, and "lower" retinal sensitivity. The variable β represents the standard regression coefficient, which when positive indicates a positive association, and when negative indicates a negative association. BCVA (logMAR) and metamorphopsia scores were positively associated, whereas retinal sensitivity was negatively associated. However, as the reviewer pointed out, this is confusing, so we added the units to Table 3 and added the following explanatory text to the Results section:

“The only factor significantly associated with BCVA at 1 year postoperatively was the presence of preoperative total retinal detachment (β=0.589, P<0.001), which was associated with significantly worse BCVA at 1 year postoperatively. The vertical metamorphopsia score on the M-chart at 1 year postoperatively was significantly related to both the presence of intraoperative ILM peeling (β=0.443, P=0.003) and the duration after recognizing visual dysfunction (β=0.425, P=0.005); these associated factors significantly worsened vertical metamorphopsia at 1 year postoperatively. The horizontal metamorphopsia score on the M-chart was related only to the duration after recognizing visual dysfunction (β=0.457, P=0.008), and a longer time to surgery was associated with significantly worse vertical and horizontal metamorphopsia scores at 1 year postoperatively. Regarding retinal sensitivity at 1 year postoperatively, only the disappearance of the EZ line on OCT at 3 months postoperatively (β=−0.638, P<0.001) was significantly associated with lower retinal sensitivity.” (lines 202–213)

Comment 6:

Among

---

## [Decision Letter · Decision Letter 1]

2 Jun 2024

PONE-D-24-04377R1Factors associated with postoperative visual function after rhegmatogenous retinal detachment with foveal detachmentPLOS ONE

Dear Dr. yokoyama,

Thank you for submitting your manuscript to PLOS ONE. After careful consideration, we feel that it has merit but does not fully meet PLOS ONE’s publication criteria as it currently stands. Therefore, we invite you to submit a revised version of the manuscript that addresses the points raised during the review process.

We look forward to receiving your revised manuscript.

Kind regards,

Jiro Kogo

Academic Editor

PLOS ONE

Reviewers' comments:

Reviewer's Responses to Questions

**Comments to the Author**

1. If the authors have adequately addressed your comments raised in a previous round of review and you feel that this manuscript is now acceptable for publication, you may indicate that here to bypass the “Comments to the Author” section, enter your conflict of interest statement in the “Confidential to Editor” section, and submit your "Accept" recommendation.

Reviewer #2: (No Response)

2. Is the manuscript technically sound, and do the data support the conclusions?

Reviewer #2: Yes

3. Has the statistical analysis been performed appropriately and rigorously? 

Reviewer #2: Yes

4. Have the authors made all data underlying the findings in their manuscript fully available?

Reviewer #2: Yes

5. Is the manuscript presented in an intelligible fashion and written in standard English?

Reviewer #2: Yes

6. Review Comments to the Author

Reviewer #2: Authors have addressed the previous reviewers’ comments well and consequently the manuscript has been revised well. However, I still find several points to be revised further mainly in the parts that have been changed by revision.

1. Abstract: “Conclusions” are written in a style like a copy of the results. Please revise the conclusions to summarize the results briefly in one or two sentences.

2. Statistical analysis: Lines 171-173 ‘including….. at 1 year postoperatively’ could be omitted because they duplicate with lines 176-177.

3. Results; Multiple regression analysis: I would like to request authors to revise whole sentences in an organized way. For example, lines 197-201 should be omitted because the same thing has been described in the methods; for each of BCVA, vertical and horizontal metamorphopsia, descriptions of ‘significantly associated with’ and ‘which was associated with significantly worse …’ should be presented in a single and simple sentence to avoid duplications; lines 204-207 might be ‘ILM peeling and longer duration were associated with higher vertical metamorphopsia score’, not ‘metamorphopsia was associated with ILM peeling and longer duration’; same in lines 208-211; ‘1 year postoperatively’ appears too often; etc.

4. 1st paragraph of discussion: Usually discussions start with a summary of the results in the current study, whereas the present writing of this paragraph sounds like an introduction to the report by Park et al. Please revise this paragraph to focus on the authors’ own results and compare them with previous literature. It might be recommended to create a new paragraph as the 1st to describe the total results briefly and move on to the 2nd in which factors related to postsurgical visual acuity are discussed.

5. Reference 18 by Abdullah et al: in this paper, all eyes were treated using both PFCL and SO, which may have had some or significant impact on the retinal function and macular morphology particularly in the ILM-peeled area as has been suggested by some previous studies and meta-analysis including the one assessing metamorphopsia and discontinuity of the ORLs between eyes treated with and without PFCL. When authors compare their results with previous literature, those difference in the details of surgical procedures that may potentially affect the results should be carefully discussed.

6. Discussion, lines 274-281: authors discuss that the discontinuity of Ez lines was recovered in many cases from 3 months to 1 year, while the Ez status at 3 months was predictive of retinal sensitivity at 1 year. To me, these two descriptions do not fit each other well and it is not clear how they are connected to each other in the structure of the paragraph.

7. In the newly added paragraph in Discussion part (from the line 296), some points are discussed very well regarding the discrepancy in postoperative metamorphopsia in comparison with previous reports. However, the first point proposed by the authors (preoperative RRD status and symptom duration) may need to show some trend that those status were not typical (if any) among their patients compared to other studies.

8. The last point in the same paragraph to note the difference in the fluid drainage techniques (lines 310-312): If this aspect affected the results, it could be interpreted that drainage from the original break using endoscopy had negative impact on the postoperative recovery of metamorphopsia and retinal sensitivity. Is it true? Personally, I do not think so because results presented in some previous literatures showed better outcomes after drainage from the original break than drainage from posterior retinotomy site, although I do not know if there is any previous report to show advantage or disadvantage when endoscopy is utilized.

7. PLOS authors have the option to publish the peer review history of their article (what does this mean?). If published, this will include your full peer review and any attached files.

Reviewer #2: No

---

## [Author Response · Author response to Decision Letter 1]

2 Jul 2024

July 2nd, 2024

Kogo Jiro 

Academic Editor

PLOS ONE

Re: Revised manuscript, PONE-D-24-04377R1

Title: Factors associated with postoperative visual function after rhegmatogenous retinal detachment with foveal detachment

Dear Dr. Kogo:

It is a pleasure to communicate with you again in relation to our abovementioned manuscript, which underwent a major revision. We have tried to amend the manuscript in accordance with the specific and general requests, and we provide below a point-by-point response to your and the reviewers’ comments. All changes in our revised manuscript are indicated by red font.

We thank you and the reviewers for your thoughtful suggestions and insights, from which our manuscript has greatly benefited. We look forward to working with you and the reviewers to move this manuscript closer to publication in PLOS ONE.

Once again, I take this opportunity to thank you, the editors, and the reviewers who we believe did an excellent editorial review job that has increased the scientific merit of this paper. Please note that we remain open to further requests for changes from you and the reviewers. 

Thank you for your consideration. I look forward to hearing from you and hope to work with PLOS ONE again in the future.

Sincerely,

Sho Yokoyama

Department of Ophthalmology, Japan Community Healthcare Organization Chukyo Hospital

1-1-10 Sanjo Minami-ku Nagoya-city, Aichi prefecture, Japan

E-mail: yokoyama@chukyogroup.jp

Reviewer #2: Authors have addressed the previous reviewers’ comments well and consequently the manuscript has been revised well. However, I still find several points to be revised further mainly in the parts that have been changed by revision.

We thank the reviewer for the thoughtful comments that have increased the scientific value of our paper. We have carefully revised the manuscript based on the reviewer’s suggestions.

Comment 1:

1. Abstract: “Conclusions” are written in a style like a copy of the results. Please revise the conclusions to summarize the results briefly in one or two sentences.

Response 1:

Thank you for this pertinent suggestion. The Conclusions section of the Abstract has been revised to provide a summary of our results.

Conclusions in Abstract:

“Our study findings suggest that BCVA, metamorphopsia, and retinal sensitivity at 1 year after vitrectomy for RRD with foveal detachment are influenced by distinct factors.” (lines 38–39)

Comment 2: 

Statistical analysis: Lines 171-173 ‘including….. at 1 year postoperatively’ could be omitted because they duplicate with lines 176-177.

Response 2:

Thank you for the valuable advice. We removed the duplicated sentences and revised them.

Statistical analysis:

“Initially, univariate linear regression analysis was performed to estimate the relationships of pre-, intra-, and postoperative factors with postoperative visual function. Subsequently, multivariate linear regression analyses were conducted using the stepwise method, incorporating factors with P-values <0.1 in the univariate linear regression analysis as explanatory variables and BCVA, vertical and horizontal metamorphopsia degree determined using M-CHARTS, and retinal sensitivity determined using the MP-3 at 1 year postoperatively as the objective variables.” (lines 164–170)

Comment 3: 

Results; Multiple regression analysis: I would like to request authors to revise whole sentences in an organized way. For example, lines 197-201 should be omitted because the same thing has been described in the methods; for each of BCVA, vertical and horizontal metamorphopsia, descriptions of ‘significantly associated with’ and ‘which was associated with significantly worse …’ should be presented in a single and simple sentence to avoid duplications; lines 204-207 might be ‘ILM peeling and longer duration were associated with higher vertical metamorphopsia score’, not ‘metamorphopsia was associated with ILM peeling and longer duration’; same in lines 208-211; ‘1 year postoperatively’ appears too often; etc.

Response 3:

Thank you for these valuable comments, and we agree with the reviewer’s suggestions. The Results (both in the Abstract and the main manuscript text) were revised accordingly.

Results in the Abstract:

“Preoperative total retinal detachment was the only factor significantly associated with worse BCVA at 1 year postoperatively (β=0.589, P<0.001). Intraoperative ILM peeling (β=0.443, P=0.003) and longer duration after recognizing visual dysfunction (β=0.425, P=0.005) were significantly associated with higher vertical metamorphopsia scores at 1 year. The horizontal metamorphopsia score was significantly related to the duration after recognizing visual dysfunction (β=0.457, P=0.008). The disappearance of the EZ line on OCT at 3 months postoperatively (β=−0.638, P<0.001) was significantly associated with lower retinal sensitivity at 1 year.” (lines 30–37)

Multiple regression analysis:

“The results of the multiple regression analysis are presented in Table 3. Preoperative total retinal detachment was the only factor significantly associated with worse BCVA at 1 year postoperatively (β=0.589, P<0.001). For vertical metamorphopsia at 1 year, intraoperative ILM peeling (β=0.443, P=0.003) and longer duration after recognizing visual dysfunction (β=0.425, P=0.005) were significantly associated with higher scores. The horizontal metamorphopsia score was significantly related to the duration after recognizing visual dysfunction (β=0.457, P=0.008), indicating that a longer time to surgery was associated with worse scores for both vertical and horizontal metamorphopsia. Additionally, the disappearance of the EZ line on OCT at 3 months postoperatively (β=−0.638, P<0.001) was significantly associated with lower retinal sensitivity at 1 year.” (lines 191–200)

Comment 4: 

1st paragraph of discussion: Usually discussions start with a summary of the results in the current study, whereas the present writing of this paragraph sounds like an introduction to the report by Park et al. Please revise this paragraph to focus on the authors’ own results and compare them with previous literature. It might be recommended to create a new paragraph as the 1st to describe the total results briefly and move on to the 2nd in which factors related to postsurgical visual acuity are discussed.

Response 4:

Thank you for this valuable suggestion. We fully agree with the reviewer’s comment and created a new first paragraph to summarize all study results.

Discussion:

“In this study investigating factors related to visual function at 1 year postoperatively, we found that preoperative total retinal detachment was significantly associated with worse BCVA after surgery. This result partially supports the findings of a previous study by Park et al. Intraoperative ILM peeling and longer duration after recognizing visual dysfunction were significantly associated with higher vertical metamorphopsia scores, and the horizontal metamorphopsia score was significantly related to the duration after recognizing visual dysfunction. Additionally, the disappearance of the EZ line on OCT at 3 months postoperatively was significantly associated with lower retinal sensitivity.” (lines 223–230)

“In the current study, total detachment emerged as a significant factor related to BCVA at 1 year postoperatively, echoing the findings by Park et al. that preoperative predictive factors include the extent of detachment.” (lines239–242)

Comment 5: 

Reference 18 by Abdullah et al: in this paper, all eyes were treated using both PFCL and SO, which may have had some or significant impact on the retinal function and macular morphology particularly in the ILM-peeled area as has been suggested by some previous studies and meta-analysis including the one assessing metamorphopsia and discontinuity of the ORLs between eyes treated with and without PFCL. When authors compare their results with previous literature, those difference in the details of surgical procedures that may potentially affect the results should be carefully discussed.

Response 5:

Thank you for these valuable remarks. We agree with the reviewer’s suggestions. As you pointed out, in the report by Abdullah et al., PFCL was used intraoperatively, and they performed SO tamponade in all patients. We have added to the manuscript that these points need to be taken into consideration, as their surgical technique differs from that of our current study, and the use of PFCL and SO tamponade may have affected both macular structure and function postoperatively.

Discussion: 

Conversely, Abdullah et al. evaluated macular structure and function with and without ILM peeling for RRD with foveal detachment using OCT angiography and multifocal electroretinography, revealing potential damage to macular structure and function in the ILM peeling group [18]. However, in the study by Abdullah et al., perfluorocarbon liquids (PFCLs) and silicone oil (SO) tamponade were used in all patients. Previous reports have indicated that the use of PFCLs is associated with increased discontinuity in the interdigitation zone following retinal detachment surgery compared with not using PFCLs [19]. Additionally, SO tamponade has been associated with postoperative thinning of both the inner and outer retinal layers compared with gas tamponade [20]. Therefore, it is important to consider the potential effects of PFCLs and SO tamponade on macular structure and function.” (lines 257–267)

19. McKay BR, Bansal A, Kryshtalskyj M, Wong DT, Berger AR, Muni RH. Two-year outcomes of different subretinal fluid drainage techniques during vitrectomy for fovea-off rhegmatogenous retinal detachments: ELLIPSOID-2 Study. Br J Ophthalmol. 2024 (online ahead of print). https://doi.org/10.1136/bjo-2023-323879. 

20. Huang RS, Mihalache A, Lau THA, Popovic MM, Kertes PJ, Muni RH. Pars plana vitrectomy with silicone oil or gas tamponade for uncomplicated retinal detachment: a systematic review and meta-analysis. Am J Ophthalmol. 2024;266: 144-155. https://doi.org/10.1016/j.ajo.2024.05.008. 

Comment 6:

Discussion, lines 274-281: authors discuss that the discontinuity of Ez lines was recovered in many cases from 3 months to 1 year, while the Ez status at 3 months was predictive of retinal sensitivity at 1 year. To me, these two descriptions do not fit each other well and it is not clear how they are connected to each other in the structure of the paragraph.

Response 6:

Thank you for pointing this out. We have added a sentence summarizing the two statements.

Discussion:

Retinal sensitivity measurements demonstrated an association with the morphology of the EZ line at 3 months postoperatively, implying that patients in whom the EZ line in the central fovea disappeared at 3 months postoperatively might experience reduced retinal sensitivity within the central 20° macular area at 1 year postoperatively. The EZ line at 3 months postoperatively was continuous in 5 patients, discontinuous in 24 patients, and absent in 4 patients. In contrast, the EZ line at 1 year postoperatively was continuous in 22 patients and discontinuous in 11 patients, with no case of disappearance, indicating an improvement in the EZ line over time after surgery. Thus, the initial state of the EZ line at 3 months is important for predicting long-term retinal sensitivity, although many patients experience recovery of the EZ line from 3 months to 1 year postoperatively.” (lines 275–284)

Comment 7:

In the newly added paragraph in Discussion part (from the line 296), some points are discussed very well regarding the discrepancy in postoperative metamorphopsia in comparison with previous reports. However, the first point proposed by the authors (preoperative RRD status and symptom duration) may need to show some trend that those status were not typical (if any) among their patients compared to other studies.

Response 7:

Thank you for your remarkable comments. In the cited article, Yamada et al. [22] reported a period from subjective symptoms to surgery of 5.25±6.37 days, which is shorter than that in the present study (8.59±21.01 days), and fewer than two quadrants were affected by retinal detachment in 27 of 33 cases, suggesting that the preoperative retinal detachment status may have been milder in their study than in the present study. In addition, the preoperative retinal detachment status was not reported by Borowicz et al. (2019) and Okuda et al. (2018). We evaluate these points in the Discussion section.

Discussion:

“First, the preoperative retinal detachment status and the duration of foveal detachment might have varied between previous studies and the current study. Yamada et al. [22] reported a period from subjective symptoms to surgery of 5.25±6.37 days, which is shorter than the 8.59±21.01 days observed in our study. Moreover, the retinal detachment area affected fewer than two quadrants in 27 out of 33 cases in Yamada et al., suggesting that the preoperative retinal detachment status in their study might have been milder than that in the current study. Additionally, the number of quadrants with retinal detachment and the duration of foveal detachment were not specified in Borowicz et al. [21] and Okuda et al. [23]. Therefore, the recovery over time might have differed among these four studies depending on the retinal detachment status of the patients.” (lines 304–313)

Comment 8:

The last point in the same paragraph to note the difference in the fluid drainage techniques (lines 310-312): If this aspect affected the results, it could be interpreted that drainage from the original break using endoscopy had negative impact on the postoperative recovery of metamorphopsia and retinal sensitivity. Is it true? Personally, I do not think so because results presented in some previous literatures showed better outcomes after drainage from the original break than drainage from posterior retinotomy site, although I do not know if there is any previous report to show advantage or disadvantage when endoscopy is utilized.

Response 8:

Thank you for your valuable feedback. The use of an endoscope allows for the drainage of subretinal fluid through existing retinal tears without the use of PFCL or the creation of a drainage retinotomy. As the reviewer pointed out, these points can result in favorable postoperative outcomes. Although no previous studies have demonstrated the usefulness of subretinal fluid drainage using an endoscope, it has been reported that avoiding the creation of a drainage retinotomy can improve the success rate of RRD surgery and reduce the incidence of postoperative ERM (Ohara et al., 2022; Ishikawa et al., 2022). Other reports implicated the use of PFCLs in postoperative visual acuity reduction and disruption of the outer retinal structures (Kumari et al., 2022). Citing these references, we have added the possibility that differences in surgical methods between previous publications and our current study may have favored the early recovery of macular function in our study.

Discussion:

“It has been reported that avoiding the creation of a retinotomy during the drainage of subretinal fluid can improve the success rate of RRD surgery and reduce the incidence of postoperative ERM [24,25]. Additionally, the use of PFCLs has been associated with a reduction in postoperative visual acuity and the disruption of the outer retinal structures [26]. These differences in surgical methods may have favored the early recovery of macular function in the present study, and at 3 months postoperatively, macular function recovery was nearly complete, with no further recovery observed over time.” (lines 321–327)

24. Ohara H, Yuasa Y, Harada Y, Hiyama T, Sadahide A, Minamoto A, et al. Drainage retinotomy is a risk factor for surgical failure after pars plana vitrectomy in patients with primary uncomplicated rhegmatogenous retinal detachment. Retina. 2022;42: 2307-2314. https://doi.org/10.1097/iae.0000000000003608.

25. Ishikawa K, Akiyama M, Mori K, Nakama T, Notomi S, Nakao S, et al. Drainage retinotomy confers risk of epiretinal membrane formation after vitrectomy for rhegmatogenous retinal detachment repair. Am J Ophthalmol. 2022;234: 20-27. https://doi.org/10.1016/j.ajo.2021.07.028. 

26. Kumari N, Surve A, Kuma

---

## [Decision Letter · Decision Letter 2]

15 Jul 2024

PONE-D-24-04377R2Factors associated with postoperative visual function after rhegmatogenous retinal detachment with foveal detachmentPLOS ONE

Dear Dr. yokoyama,

Thank you for submitting your manuscript to PLOS ONE. After careful consideration, we feel that it has merit but does not fully meet PLOS ONE’s publication criteria as it currently stands. Therefore, we invite you to submit a revised version of the manuscript that addresses the points raised during the review process.

We look forward to receiving your revised manuscript.

Kind regards,

Jiro Kogo

Academic Editor

PLOS ONE

Journal Requirements:

Reviewers' comments:

Reviewer's Responses to Questions

**Comments to the Author**

1. If the authors have adequately addressed your comments raised in a previous round of review and you feel that this manuscript is now acceptable for publication, you may indicate that here to bypass the “Comments to the Author” section, enter your conflict of interest statement in the “Confidential to Editor” section, and submit your "Accept" recommendation.

Reviewer #2: (No Response)

2. Is the manuscript technically sound, and do the data support the conclusions?

Reviewer #2: Yes

3. Has the statistical analysis been performed appropriately and rigorously? 

Reviewer #2: Yes

4. Have the authors made all data underlying the findings in their manuscript fully available?

Reviewer #2: Yes

5. Is the manuscript presented in an intelligible fashion and written in standard English?

Reviewer #2: Yes

6. Review Comments to the Author

**Reviewer #2:** Authors have addressed the former reviewers’ comments well, while further problems may have arisen in association with the revised parts.

1. The first paragraph of Discussion is very long and does not focus on a single theme. As pointed out in the previous review, please present a brief summary of the current study in the first paragraph (never go beyond it), and then discuss each of the details in comparison with previous literatures in the following paragraphs. Please stick to the rule of writing to focus on one particular topic in each paragraph.

2. Lines 242-243: In which study were “duration after recognizing visual dysfunction and the postoperative status of the EZ line not identified as associated factors? In the current study, those factors were shown to be associated with visual outcomes as illustrated in Table 3, then were they not identified in Park’s report? Please note this point clearly. This confusion occurs partly due to the inappropriate configuration of the paragraphs as suggested above.

3. Lines 260-270: The newly added sentences (lines 260-267; seems too long and interferes the flow of the discussion) and the next sentence (“The findings of the current study also indicate that ….”) do not logically connected to each other smoothly. The structure of discussion should be re-considered for this paragraph.

7. PLOS authors have the option to publish the peer review history of their article (what does this mean?). If published, this will include your full peer review and any attached files.

Reviewer #2: No

---

## [Author Response · Author response to Decision Letter 2]

30 Jul 2024

July 30th, 2024

Kogo Jiro 

Academic Editor

PLOS ONE

Re: Revised manuscript, PONE-D-24-04377R2

Title: Factors associated with postoperative visual function after rhegmatogenous retinal detachment with foveal detachment

Dear Dr. Kogo:

It is a pleasure to communicate with you again in relation to our abovementioned manuscript, which underwent a minor revision. We have tried to amend the manuscript in accordance with the specific and general requests, and we provide below a point-by-point response to your and the reviewers’ comments. All changes in our revised manuscript are indicated by red font.

We thank you and the reviewers for your thoughtful suggestions and insights, from which our manuscript has greatly benefited. We look forward to working with you and the reviewers to move this manuscript closer to publication in PLOS ONE.

Once again, I take this opportunity to thank you, the editors, and the reviewers who we believe did an excellent editorial review job that has increased the scientific merit of this paper. Please note that we remain open to further requests for changes from you and the reviewers. 

Thank you for your consideration. I look forward to hearing from you and hope to work with PLOS ONE again in the future.

Sincerely,

Sho Yokoyama

Department of Ophthalmology, Japan Community Healthcare Organization Chukyo Hospital

1-1-10 Sanjo Minami-ku Nagoya-city, Aichi prefecture, Japan

E-mail: yokoyama@chukyogroup.jp

Reviewer #2: Authors have addressed the previous reviewers’ comments well and consequently the manuscript has been revised well. However, I still find several points to be revised further mainly in the parts that have been changed by revision.

We thank the reviewer for the thoughtful comments that have increased the scientific value of our paper. We have carefully revised the manuscript based on the reviewer’s suggestions.

Comment 1:

The first paragraph of the Discussion is very long and does not focus on a single theme. As pointed out in the previous review, please present a brief summary of the current study in the first paragraph (never go beyond it), and then discuss each of the details in comparison with previous literatures in the following paragraphs. Please stick to the rule of writing to focus on one particular topic in each paragraph.

Response 1:

Thank you for your insightful suggestion. We have revised the first paragraph of the Discussion section as the reviewer recommended. The first paragraph now provides a summary of the current study, while the subsequent paragraphs discuss each detail in comparison with previous literature.

“In this study investigating factors related to visual function at 1-year postoperatively, we found that preoperative total retinal detachment was significantly associated with worse BCVA after surgery. Intraoperative ILM peeling and longer duration after recognizing visual dysfunction were significantly associated with higher vertical metamorphopsia scores, and the horizontal metamorphopsia score was significantly related to the duration after recognizing visual dysfunction. Additionally, the disappearance of the EZ line on OCT at 3 months postoperatively was significantly associated with lower retinal sensitivity.” (lines 224-230)

Comment 2:

Lines 242-243: In which study were “duration after recognizing visual dysfunction and the postoperative status of the EZ line not identified as associated factors? In the current study, those factors were shown to be associated with visual outcomes as illustrated in Table 3, then were they not identified in Park’s report? Please note this point clearly. This confusion occurs partly due to the inappropriate configuration of the paragraphs as suggested above.

Response 2:

Thank you for your question.

In our study, we did not identify “duration after recognizing visual dysfunction and the postoperative status of the EZ line” as factors associated with BCVA in the first postoperative year. On the other hand, Park et al. identified the extent of detachment; duration of macula-off; preoperative external limiting membrane (ELM) integrity; and postoperative outer retinal microstructures, particularly the photoreceptor outer segment layer and postoperative microstructure of the outer retina, as factors associated with visual acuity in the first year. We have revised the text as follows to clarify the differences between this study and previous reports, including Park et al.

“Park et al. examined factors influencing visual acuity 1-year postoperatively in 180 eyes undergoing surgery for RRD with foveal involvement. They reported that predictive factors included the extent of detachment, duration of macula-off, preoperative external limiting membrane (ELM) integrity, and postoperative outer retinal microstructures, particularly the photoreceptor outer segment layer [6]. Additionally, previous research has emphasized the significance of OCT morphology after RRD surgery, indicating that the continuity of the postoperative EZ and ELM lines, as well as the presence of a foveal bulge, correlate with final BCVA after RRD surgery [12–14].

On the other hand, our study identified total detachment as a significant factor related to BCVA 1-year postoperatively, aligning with Park et al.'s findings that preoperative predictive factors include the extent of detachment. However, our study did not identify the duration after recognizing visual dysfunction or the postoperative status of the EZ line as associated factors. The discrepancy between previous reports of the duration of macula-off and the OCT morphology after RRD surgery being associated with postoperative visual acuity and our results may be attributed to the relatively small sample size in our study and the less stringent evaluation of the duration of foveal detachment.” (lines 231-247)

Comment 3:

Lines 260-270: The newly added sentences (lines 260-267; seems too long and interferes the flow of the discussion) and the next sentence (“The findings of the current study also indicate that ….”) do not logically connected to each other smoothly. The structure of discussion should be re-considered for this paragraph.27–32. doi: 10.1097/iae.0000000000003259. 

Response 3:

Thank you for your valuable suggestion. We have made the following changes to the structure of the text in the Discussion section.

“The duration after recognizing visual dysfunction was identified as an associated factor for both vertical and horizontal metamorphopsia scores assessed using M-CHARTS. Although the duration after recognizing visual dysfunction may not always coincide with the period of foveal detachment, these two intervals are considered similar owing to the impact of central foveal detachment on visual acuity. Thereby, the current study results suggest that postoperative metamorphopsia may worsen with a longer duration since the patient acknowledged central vision dysfunction.

Additionally, intraoperative ILM peeling was associated with worse postoperative 1-year vertical metamorphopsia scores. In the context of vitrectomy for patients with RRD, ILM peeling may be performed to prevent postoperative ERMs [15,16]. In regard to postoperative retinal function with ILM peeling in RRD cases, Pietras-Trzpiel et al. reported no significant difference in postoperative metamorphopsia scores assessed using M-CHARTS between patients with or without ILM peeling during vitrectomy for RRD [17]. Conversely, Abdullah et al. evaluated macular structure and function with and without ILM peeling for RRD with foveal detachment using OCT angiography and multifocal electroretinography, revealing potential damage to macular structure and function in the ILM peeling group [18]. However, in that study, perfluorocarbon liquids (PFCLs) and silicone oil (SO) tamponade were used in all patients. Reportedly, the use of PFCLs is associated with increased discontinuity in the interdigitation zone following RRD surgery compared with that when not using PFCLs [19]. Additionally, SO tamponade has been associated with postoperative thinning of both the inner and outer retinal layers compared with that using gas tamponade [20]. Therefore, it is important to consider the potential effects of PFCLs and SO tamponade on macular structure and function.

In the current study, the decision to perform ILM peeling was left to the discretion of individual surgeons, and it is possible that more severe cases were more likely to undergo ILM peeling. Indeed, the higher the preoperative PVR grade, the more frequently ILM peeling was performed. However, the results of multiple regression analysis, which included the presence of ILM peeling and PVR grade as factors, showed that PVR grade was not related to postoperative visual function, including visual acuity, metamorphopsia, and retinal sensitivity. In summary, the current study suggests that ILM peeling may impact retinal function and exacerbate metamorphopsia regardless of the severity of the PVR grades.“ (lines 248-277)

---

## [Editor Report · Decision Letter 3]

1 Aug 2024

Factors associated with postoperative visual function after rhegmatogenous retinal detachment with foveal detachment

PONE-D-24-04377R3

Dear Dr. Yokoyama

We’re pleased to inform you that your manuscript has been judged scientifically suitable for publication and will be formally accepted for publication once it meets all outstanding technical requirements.

Kind regards,

Jiro Kogo

Academic Editor

PLOS ONE
---

## [Editor Report · Acceptance letter]

5 Aug 2024

PONE-D-24-04377R3 

PLOS ONE

Dear Dr. yokoyama, 

I'm pleased to inform you that your manuscript has been deemed suitable for publication in PLOS ONE. Congratulations! Your manuscript is now being handed over to our production team.

Kind regards, 

on behalf of

Prof. Jiro Kogo 

Academic Editor

PLOS ONE